# Usual care in a multicentre randomised controlled trial of financial incentives for smoking cessation in pregnancy: qualitative findings from a mixed-methods process evaluation

Jennifer McKell ,[1] Fiona M Harris ,[2] Lesley Sinclair ,[3] Linda Bauld ,[4] David Michael Tappin ,[5] Pat Hoddinott [6]

For numbered affiliations see end of article.

**Correspondence to**
Jennifer McKell;
j.e.mckell@stir.ac.uk

## ABSTRACT

**Objectives** Financial incentives are recommended by the UK's National Institute for Health and Care Excellence to aid smoking cessation in pregnancy. However, little is known about how implementation contexts might impact on their effectiveness. Variations in smoking cessation support (usual care) for pregnant women who smoke were examined qualitatively as part of a prospective process evaluation of the Cessation in Pregnancy Incentives Trial (CPIT III).

**Design** Longitudinal case studies of five CPIT III trial sites informed by realist evaluation.

**Setting** A stop smoking service (SSS) serving a maternity hospital constituted each case study, located in three UK countries.

**Participants** Data collection included semistructured interviews with trial participants (n=22), maternity (n=12) and SSS staff (n=17); and site observations and perspectives recorded in fieldnotes (n=85).

**Results** Cessation support (usual care) for pregnant women varied in amount, location, staff capacity, flexibility and content across sites. SSS staff capacity was important to avoid gaps in support. Colocation and good working relationships between maternity and SSS professionals enabled prioritisation and reinforced the importance of smoking cessation. Sites with limited use of carbon monoxide (CO) monitoring reduced opportunities to identify smokers while inconsistency around automatic referral processes prevented the offer of cessation support. SSS professionals colocated within antenatal clinics were available to women they could not otherwise reach. Flexibility around location, timing and tailoring of approaches for support, facilitated initial and sustained engagement and reduced the burden on women.

**Conclusions** Trial sites faced varied barriers and facilitators to delivering cessation support, reflecting heterogeneity in usual care. If financial incentives are more effective with concurrent smoking cessation support, sites with fewer barriers and more facilitators regarding this support would be expected to have more promising trial outcomes. Future reporting of trial outcomes will assist in understanding incentives' generalisability across a wide range of usual care settings.

**Trial registration number** ISRCTN15236311.

## STRENGTHS AND LIMITATIONS OF THIS STUDY

⇒ This study uniquely examines, in depth, usual care provided during a multicentre trial of financial incentives for smoking cessation in pregnancy.
⇒ Observations of usual care delivery, and environment, plus interviews with trial participants and healthcare staff allowed analysis of multiple perspectives on usual care and trial impact.
⇒ Anonymous presentation of case study data allows for a more candid discussion of barriers and facilitators within sites.
⇒ Case study sites represented populations with a majority white British background therefore barriers and facilitators to cessation support may differ in other more diverse populations.
⇒ Barriers and facilitators relating to communication skills, such as motivational techniques used by stop smoking service professionals to support cessation in pregnant women, were not considered.

## INTRODUCTION

Smoking in pregnancy raises the risk of serious consequences for mothers and babies, including miscarriage, stillbirth, prematurity, birth defects and infant death.[1] Women who continue to smoke can lose on average 10 years of life, whereas those who give up permanently by age 40 during their childbearing years are likely to have a normal lifespan.[2] Identifying effective and cost-effective ways to help more women to quit smoking during pregnancy remains vital, and evidence supports the use of financial incentives.[3] In 2021, National Institute for Health and Care Excellence (NICE) recommended the use of financial incentives to help pregnant women to quit smoking.[4]

Alongside effectiveness and cost effectiveness for behavioural interventions, it is important to understand how complex

interventions work.[5] Support received by pregnant women alongside incentives may be a factor influencing effectiveness.[6] A pilot study in Scotland found that participants most successful in their quit attempts were those who perceived financial incentives as 'part of a wider reward structure' involving regular engagement with specialist support.[7] Similarly, Mantzari *et al* found that cessation support held greater importance for incentivised compared with non-incentivised women, with the former particularly motivated by 'being monitored'.[8] This evidence followed the publication of NICE guidelines in 2010, recommending 'intensive and ongoing support' of pregnant women by National Health Service (NHS) smoking cessation services as part of usual care.[9] All pregnant women resident in the UK are eligible to receive free maternity care provided by the NHS. There are differences in how care is provided across the devolved UK nations,[10–12] but generally this involves a combination of hospital and community-based care. Smoking cessation support within pregnancy, in the UK, varies from one area to the next but particular characteristics occur regularly.[13]

Offering smoking cessation support alongside financial incentives was a key characteristic of the single-centre Cessation in Pregnancy Incentives Trial (CPIT II).[14] Participants receiving the addition of incentives were two and a half times more likely to be abstinent at the end of pregnancy compared with SSS support alone (22.5%, 69/306; vs 8.6%, 26/303). The CPIT III trial commenced in 2018, replicating the CPIT II intervention and trial design in seven sites across three UK countries. CPIT III aimed to establish, the effectiveness, cost effectiveness and generalisability of adding the offer of financial incentives to usual care to stop smoking during pregnancy.[13] Like the findings of CPIT II, CPIT III found that participants receiving incentives in addition to usual care were more than twice as likely to be smoke free in late pregnancy compared with participants receiving usual care alone (26.8%, 126/471; vs 12.3%, 58/470).[15]

The trial protocol[13] included a mixed-methods process evaluation to identify and understand differences in usual care for pregnant smokers; the geographic and population mix of trial sites and their relationship to trial conduct and outcomes.

This paper sets out the barriers and facilitators to providing SSS support (usual care) at five of seven trial sites. An a-priori proposition underpinned the process evaluation:

Sites would vary in the usual care provided for pregnant women who smoke and therefore trial outcomes would vary by site, indicating that maternity services and SSSs interact with the offer of financial incentives.

## METHODS
### Process evaluation design
A longitudinal, mixed-methods case study design, informed by realist evaluation, formed the basis of the embedded process evaluation.[5 16 17] The case studies, defined as a trial site (an SSS serving a maternity hospital), included five of the seven trial sites representing diverse service configurations. Being one of the first five sites to open to recruitment was the criteria for selection of case study sites, though the fifth site chosen opened later than expected. All case study sites were selected independent of the main trial team. Methods and findings described in this paper relate to qualitative examination of usual care. Further details on the design of the CPIT III full process evaluation are reported elsewhere.[13]

### Patient and public involvement
Two CPIT II participants and members of the UK Centre for Tobacco and Alcohol Studies (UKCTAS) smokers' panel were involved in planning CPIT III; and a patient representative was included in the CPIT III trial steering committee. Resources were not available to provide separate patient and public involvement in the process evaluation and the onset of the COVID-19 pandemic towards the end of the study, and during analytical and interpretative stages, further restricted opportunities for further involvement.

### Data collection
Case study data collection occurred in two stages. Stage 1 (October 2017 to December 2019) established the context for the trial locally via observation of usual care of pregnant smokers and informal interviews with maternity or SSS staff recorded as fieldnotes. Stage 2 (July 2018 to September 2020) involved qualitative, semistructured, interviews using predetermined topic guides (online supplemental files 1 and 2) with trial participants, maternity and SSS staff (healthcare staff) to explore experiences and perceptions relating to the trial and usual care. Concurrent to stages 1 and 2, the process evaluation team recorded, also in fieldnotes, trial staff perspectives relating to case study sites from trial management working group (n=31) meetings and informal discussions with the trial manager (n=6) and site-based trial staff (n=16). Case study-specific data sources are detailed in table 1. To protect the anonymity of interviewees and informants, case study sites are referred to as A–E.

Data collection was undertaken by JM with assistance from IU (please see Acknowledgements) who are both female qualitative researchers with experience of research into smoking cessation and pregnant women. Interviews with trial participants were conducted by telephone and lasted on average approximately 20 min, with duration ranging from 9 to 36 min. Sample sizes at sites were proportionate to the numbers recruited and aimed to achieve maximum diversity in participant age, ethnicity and socioeconomic characteristics in sites with higher numbers. Two women who consented to contact about the trial but did not consent to be randomised were interviewed but elicited little additional data. Interviews with healthcare staff were conducted either face to face (n=19) or via telephone (n=10) and ranged from 11 to 57 min in duration, or on average 30 min. Face-to-face interviews were conducted in workplaces (apart from one in a public

**Table 1** Case study-specific data sources (n=5)

| Data source | Site A | Site B | Site C | Site D | Site E | Total |
|---|---|---|---|---|---|---|
| Observations of maternity booking or other early antenatal appointments | 2 | 3* | 2 | 5 | 3 | 15 |
| Observations of smoking cessation consultations | 3 | 3 | 2 | 2 | 2 | 12 |
| Informal interviews with maternity or SSS staff | 1 | 1 | 1 | 1 | 1 | 5 |
| Interviews with maternity staff | 2 | 4† | 3 | 3 | 0‡ | 12 |
| Interviews with SSS staff | 3† | 3 | 3 | 5 | 3 | 17 |
| Interviews with trial participants | 2 | 4§ | 6 | 5 | 5 | 22 |
| Informal discussions with local trial team members (number of trial team members consulted) | 3 (1) | 2 (1) | 5¶(3) | 5 (2) | 1 (1) | 16 |
| Total | 16 | 20 | 22 | 26 | 15 | 99 |

*Includes observation of a training session for midwives delivered by SSS.
†One interview included two interviewees.
‡It was not possible to conduct interviews with maternity staff due to the onset of the Covid-19 pandemic and ongoing pressures up to the end of qualitative data collection in December 2020.
§Includes 2 interviewees who indicated an interest in the study but were not randomised.
¶Includes one semistructured formal interview with a local research lead, conducted by telephone. Two other trial team members consulted four times.
SSS, stop smoking service.

place). Average duration of face-to-face interviews was similar to that of telephone interviews, at around 36 min. Two, face-to-face healthcare staff interviews involved the participation of two interviewees in each; the remaining interviews had no other people present. All interview participants were unknown to researchers prior to study commencement, and information provided before interviews explained the researcher's role in the research.

## Analysis

Analysis of data drew on the Framework method[18 19] informed by realist evaluation. Transcripts of interviews with healthcare staff and field notes from site observations, informal interviews with healthcare staff, discussions with trial staff and trial meetings were uploaded to QSR Nvivo (V.12) software for purposes of organisation and management. Informed by themes in the topic guides but also themes arising from the data, an initial coding framework was created, tested and developed by JM and IU to produce a finalised version (online supplemental file 3). This was then applied to the data in the Nvivo file as part of the indexing stage. Transcripts of trial participant interviews were read and indexed separately according to two themes: perspectives of usual care and trial processes, respectively. As part of the charting phase, matrices were developed for each case study site containing data summaries incorporating the theme of 'routine service characteristics' from the coding framework and perspectives of usual care from trial participant interviews. Barriers and facilitators to cessation support (usual care) were then drawn out by comparing the data summaries to identify similarities and contradictions in the data. Identified barriers and facilitators were considered in depth to establish Context, Mechanism Outcome configurations (CMO)

CMOs were developed with input from the wider process evaluation team (PH and FH), which included both social science and clinical input. The 'a priori' proposition was then revisited in the light of our findings, to inform the discussion.

Participant confidentiality was important to facilitate honest and information-rich accounts of usual care. Staff job titles within SSSs significantly varied across the sites, often reflecting their occupations. In some sites, those providing cessation support were specialist midwives, while in another, nurses delivered cessation support. In two sites, support was delivered by 'facilitators' or 'well-being workers', with no clinical background. To minimise jigsaw identification of people or sites, quotes from SSS staff that deliver cessation support are referred to using the generic term 'SSS professional', regardless of occupational background, and 'senior' used for individuals with line management responsibility. Any potentially identifying information has been removed.

## RESULTS

Usual care settings varied considerably (table 2). Barriers and facilitators to delivering SSS support during a financial incentives trial are then presented, with CMO configurations to highlight the implications for usual care.

### Barriers and facilitators to providing usual care in trial sites

Case studies highlighted barriers and facilitators to providing usual care, particularly in relation to professional relationships and service capacity. Barriers and facilitators were also identified in maternity services' ability to identify smoking in pregnancy and SSS ability to engage women in cessation support, as well as features of

**Table 2** Usual care settings in CPIT III trial sites

| | Site A | Site B | Site C | Site D | Site E |
|---|---|---|---|---|---|
| Policy of referral (opt-in or opt-out) | Opt-out | Opt-out | Opt-out | Opt-out | Opt-out |
| Target population and behaviour change supported | Pregnant women, smoking and weight management | Pregnant women, smoking only | Pregnant women, smoking only | General population, smoking only | General population, smoking, alcohol, weight management and physical activity |
| Organisation providing SSS* | NHS | NHS | NHS | NHS | Local Government |
| SSS professionals with or without a midwifery/nursing background | Mixed—some with and some without a midwifery/ nursing background | Midwifery/nursing background | Midwifery/nursing background | Midwifery/nursing background | No midwifery/ nursing background |
| SSS funder (and location of line management) | Local Government (NHS Trust/Board) | NHS Trust/Board (NHS Trust/Board) | Public Health Body (NHS Health Improvement) | NHS Health Improvement (NHS Health Improvement) | Local Government (Local Government) |
| Venue for SSS consultations | Hospital consultation room or other space (not dedicated). Also, a GP Surgery room for those remotely located | Hospital consultation room (dedicated) or home visit | Hospital consultation room (dedicated) | Community venue (regular drop-in or by appointment) or home visit | Community venue (regular drop-in or by appointment) |
| Methods of SSS consultation | Face-to-face or telephone | Face-to-face or telephone | Face-to-face or telephone | Face-to-face or telephone | Face-to-face or telephone |
| SSS ability to provide NRT directly | No | Yes | No | Yes | No |
| Colocation of SSS and maternity services | Yes | Yes | Yes | No (in most cases) | No |

*SSS has been used to denote stop smoking services within trial sites, although some were not necessarily known as such, as they provided wider behaviour change services.
GP, general practitioner; NHS, National Health Service; NRT, nicotine replacement therapy; SSS, stop smoking service.

that support, as described below. Each section concludes with the CMO configuration.

### SSS capacity

Limitations on SSS staff capacity (number, workload, absence cover and unfilled vacancies) in trial sites affected cessation support. In site C, capacity issues arose during periods of long-term leave among key staff in a small team who were not quickly replaced or on the same number of hours. An intervention participant noted the impact of this staff shortage saying she would have liked more support from the study team.

> I wasn't getting enough phone calls I don't think, […], obviously because I wasn't getting the [SSS professional] support I would have liked you know a wee phone call just, not like a formal phone call to say oh you are going to get a voucher, but just like a wee phone call to say 'oh how are you doing?;[…]', I know that's not like your job or whatever but I

think that would have made it a wee bit more easier. (Intervention participant, Site C)

The larger SSS in site D experienced numerous staff leaving during the study period and slow recruitment to posts. However, these difficulties were not reflected in participant or SSS professionals' perspectives. Yet evidence suggested that the service may be struggling with capacity. Trial staff noticed increasing numbers of participants being referred to local pharmacies for support, plus inconsistent CO monitoring, as per the CPIT III trial protocol,[13] in intervention participants to verify smoking abstinence 4 weeks after setting a quit date.

Staff capacity issues were less prominent in other sites but a SSS manager in site E (supporting multiple behaviour change) noted that staff numbers had reduced by more than half despite increasing demand:

> [name of SSS] […] at the time, I think, had 41.5 full-time equivalent employees. So, we're talking about a vast big service. I think it was like £1.1 million pounds

the service cost and now I think we've got 16.5 full-time equivalents, (…). So, we've slowly got smaller and smaller, and I would say that the demand on our service is far, far, higher than it ever was when we were 41.5 equivalent employees. (Senior SSS professional, Site E)

Problems with capacity in SSSs (C) reduced the time and attention staff could provide (M), increasing the likelihood of suboptimal cessation support for pregnant women (O).

### Communication and connections between midwives and SSS professionals

Shared working environments and positive, constructive relations between maternity services and SSSs were beneficial. Midwives and SSS professionals in three sites were colocated in the same hospital, often within easy reach, and such proximity allowed development of close, amiable working relationships. In site B, the SSS was embedded within the maternity unit, as demonstrated by efforts to extend CO monitoring in midwife appointments:

…when we brought in carbon monoxide and every opportunity we see them … everyone [midwives] initially thinks: 'oh my God that is extra work' and we go: 'well you are already doing their blood pressure and dipping their urine so by the time you've done that, you know, it won't take you that much longer', and everyone has just got used to it… they sort of, are used to us saying oh we are doing this now and explaining it… (SSS professional, Site B)

Proximity working, however, was no guarantee of good working relationships. Midwives and SSS staff in one site shared consultation rooms and corridors, but gaps existed between them in relation to approaches to smoking cessation, joint working and communication. A senior SSS staff member highlighted mismatched priorities between maternity and SSS staff.

Community midwives what they say…yes there are some that are still perhaps not presenting it in the way that would benefit (…) the women are saying 'oh no I don't want to quit. I don't want to see them' and rather than perhaps being more positive about the [SSS] service and saying they'll have a good chat with you about the benefits of quitting or whatever. They are just informing us that these women don't want to be approached. (Senior SSS professional, Site A)

In two sites, maternity and SSS staff never or only occasionally shared the same work site, despite both working within community settings. The extent of links between staff groupings was sufficient for most but one SSS professional said they would like midwives' support to engage pregnant smokers.

if we could go and see them [midwives] and say look that girl hasn't turned up and she's coming back, if we can't get her on the phone, it might be helpful if they speak to them then. (…) So that they're getting the advice from, at each contact basically (…) (SSS professional, Site D)

Colocation and good working relationships between maternity and smoking cessation professionals (C) prioritises and reinforces smoking cessation (M) optimising the importance of quitting for women (O).

### Identifying pregnant smokers

Identification of smokers took place during initial appointments with midwives. Women were asked if they smoked and to provide a CO breath test. This was requested of all pregnant women, and not only those who self-reported smoking, to detect CO exposure and provided further opportunity to identify smokers. All sites operated an opt-out referral policy which meant that pregnant women who self-reported as smokers/recent ex-smokers or exceeded a particular CO level were automatically referred to a SSS.

The role and level of implementation of CO monitoring varied by site. There was clear expectation at some sites that midwives would undertake CO monitoring at every antenatal appointment, whereas in others, CO monitoring only occurred at initial appointments. In most sites, exceeding a particular CO level, usually of four parts per million or above, triggered referral to a SSS for both self-reported non-smokers and smokers. In contrast, at one site, self-reported non-smokers with high CO levels were not referred but recommended for future monitoring. This suggests the following CMO configuration.

Limitations on the role and use of CO monitoring within sites (C) reduced the opportunities for midwives to identify smoking in pregnancy (M) and in turn reduced opportunities to offer cessation support (O).

Differences in interpretation of automatic referral were evident among midwives, suggesting inconsistency in opportunities for smoking cessation support. Discussion of automatic, opt-out referral with staff across and within sites provided contrasting descriptions of the conditions for this. Perceptions included automatic referral not taking place if a woman particularly objected versus automatic referral conditional on specified criteria alone:

I haven't referred anybody against their will kind of thing, you know? We do try to treat it as more of a kind of opt out service so (…) So it's more they would need to kind of say, you know what I mean, instead of us just offering would you like this or not. (Midwife, Site D)

It's a mandatory referral at booking, (…) I always say you will get a phone call and it's their choice then as to whether they do, because we refer everybody. Sometimes a woman will say 'I am definitely not going to be engaging with them'. And then on the referral we will put (…) 'does not want to engage', so that's the end of it really but at least the team know about them. (Midwife, Site B)

This suggests: adherence to automatic referral based on specified criteria (C) ensures that all women are approached and informed about availability of a SSS (M) and raises the possibility of SSS engagement (O).

### Engagement in cessation support

SSSs used multiple methods to engage pregnant smokers following maternity referral, including phone-call, text and letter but cessation staff acknowledged that often women could not be reached or were lost to follow-up. Access to antenatal clinics, however, gave cessation staff an advantage in contacting women. Staff who worked within, or close to, antenatal clinics benefitted from opportunities to approach women in person. However, there was no indication in the data that a face-to-face approach was important for establishing a continuing relationship.

> …we are fortunate here in that we see the majority of women coming through the clinic either here or at one of the outlying clinics and for their initial scan. And so, if they're not picking the phone up to us then…we will more than likely see them when they attend for their scan and be able to have a discussion with them about smoking in pregnancy and whether they are ready to quit. (Senior SSS professional, Site A)

Sites that enabled SSS professionals to approach women face to face at ante-natal clinics (C) provided a sense of immediacy for smoking cessation support and reduced the potential for avoidance (M), increasing the opportunities to offer support (O)

SSS' ability to tailor the offer of support to optimise appeal to pregnant women also appeared important for engagement. In two sites, the SSS saw opportunities in the way that smoking in pregnancy is discussed. Here staff advocated for initial conversations to emphasise exposure to CO rather than stopping smoking. A SSS professional in site C described a tailored approach for reluctant referrals.

> it's just getting the wording right, (…) sometimes I would talk about you know, 'you had your booking appointment the other day, how did you get on?, Was your scan okay?, great, I see one of your tests, your CO level was very high, it was into the red there, we'd be a bit worried there for your pregnancy, you are a smoker is that right?' (…) it's not like: [SSS professional]: 'I am here to help you stop smoking'; [hypothetical woman]: 'well, I don't want to stop smoking", hang up the phone'. (SSS professional, Site C)

At site D, the SSS were unable to engage a trial participant allocated to incentives who declined support due to her perception of available support. The woman felt self-conscious of being a young pregnant smoker attending a pharmacy or a 'group': 'some people are too embarrassed to go to the groups'; though the latter was an old SSS term for a drop-in session.

SSS' ability to customise their offer of cessation support (C) to maximise appeal to individuals with different perspectives and preferences (M) facilitated initial engagement with cessation support (O).

### Features of cessation support

SSS' ability to offer flexibility regarding consultations at suitable times and convenient locations made it easier for women to engage. Participants spoke of the value of a SSS provided near to where they lived or via home visits.

> I can phone her [SSS professional] if I'm stuck at any point. But the good thing is as well is if there's a time that I'm unwell she can come out to me, I just phone her. (Intervention Group Participant, Site D)

However, SSSs were not always able to provide convenient access. An SSS located in a maternity unit did not have a dedicated consulting room and could only offer afternoon appointments after antenatal clinics. Travelling into the maternity unit was also a challenge for women living in more remote areas.

> …certainly with the follow-ups because those appointments aren't quite so long, so women are quite reluctant to get on a bus for 45 minutes to have…to see one of us for 20 minutes. (Senior SSS professional, Site A)

Site E's SSS was able to offer weekly drop-in sessions in a community centre, one of several service venues, but consultations were limited to 15 min. The service manager noted less flexibility over the location and timing of appointments if a longer one-to-one consultation was required.

Providing cessation support in accessible locations and/or at convenient times (C) facilitated engagement and minimised the commitment required from women (M) increasing opportunities for sustained engagement in support (O)

How nicotine replacement therapy (NRT) was provided also had an effect on outcomes. A lack of directly provided NRT in some SSSs caused delays to quit attempts. While some SSSs were able to provide NRT directly, others had to refer women to their general practitioner (GP) or linked pharmacies for NRT.

> So, we direct the email, the prescription request, to the doctor's surgery to guarantee it gets there. (…) We used to give them [women] it and they'd be like oh we didn't' make it to the chemist and they've still got it a week later and stuff like that. […] They can pick that up within forty-eight hours, but we do put urgent, so we can phone the doctors and ask if it can be quicker, if they want to try to get that quit date set quicker. (SSS professional, Site A)

Provision of NRT at smoking cessation consultations (C) ensures immediate access to pharmacological support (M) enhancing pregnant smokers' ability to address nicotine withdrawal (O).

Some SSSs showed a willingness to adapt to women's needs and preferences through innovation around support. Examples included extra support when dealing with low mood and depression; suggesting the use of e-cigarettes and introducing cessation support for significant others.

> …we realised that one thing that we were coming up against quite a lot was problems with mental health and general wellbeing. So, we did an audit of our patients that were being referred into us and 50% to 60% of women that were being referred into our service had either mental health issues or general wellbeing issues […] we also run a class with [charity] […] So it all talks about mental health and antenatal depression, postnatal depression, things to look out for… (SSS professional, Site C)

SSS willingness to innovate (C) extended the variety of support options available to women (M) increasing opportunities to identify effective cessation support (O).

## DISCUSSION
### Main findings
Cessation support provided as usual care by maternity services and SSSs across trial sites varied substantially in terms of SSS staff capacity and relationships, how services identified pregnant women who smoke and their ability to engage, support and help women to quit.

Limitations on staff capacity in particular sites affected the amount and type of cessation support available to women. Close contact and good relationships between midwives and SSSs were generally beneficial, but in some sites, there was only limited contact and cohesiveness.

CO monitoring of pregnant women throughout pregnancy was standard procedure in some sites but not others and this limited identification. Automatic opt-out referral of women who were smoking or exceeded a CO threshold was official policy across all sites but differences in interpretation meant that some women who smoked were not referred. Working within or close to antenatal clinics enabled SSSs to approach women they may otherwise struggle to reach. SSSs found benefit in tailoring their approach to take account of women's perspectives and preferences. Women valued cessation support consultations held in accessible locations at a convenient time of day, but some SSSs were unable to offer this flexibility. Delays to women commencing quit attempts were identified when SSSs could not provide NRT directly. Willingness and capacity by some sites to offer innovative approaches provided women with a greater variety of options and facilitated engagement with support.

### Comparison of findings with other studies
This process evaluation is unique in detailing site variation in usual care within a randomised controlled trial of financial incentives for smoking cessation in pregnancy. Usual care was described as variable in a previous cluster randomised controlled trial of a self-help smoking cessation intervention for pregnant women, however no detail was provided, and the intervention was found to be ineffective.[20] An earlier systematic review of additional support provided alongside financial incentives for smoking cessation in pregnancy highlighted the potential importance of this as contributing to effectiveness.[6] An implementation study that examined the introduction of BabyClear©, an intervention involving universal CO monitoring in maternity booking appointments and an opt-out referral pathway, found that smoking cessation referrals and the proportion of women who had quit smoking by the time of birth in implementation sites increased.[21] A process evaluation identified barriers and facilitators to implementation that echoed findings in our study. As well as the benefits of partnership working between midwives and SSS staff, and flexible appointments, it also found inconsistency in CO monitoring.[22 23] In contrast to CPIT III, the authors identified midwives' reservations around discussing smoking with pregnant women. Other qualitative research similarly highlighted midwives' concerns about feeling poorly equipped to discuss smoking in pregnancy, but also differences among midwives in perspectives of opt-out referral, as discovered during CPIT III.[24] The same study also found barriers to good working relationships between midwives and SSS staff in terms of level of contact and prioritisation of smoking in pregnancy. A study of the development and implementation of an intervention to help young pregnant women to quit smoking found that home visits, sensitivity to inequalities and direct provision of NRT were important facilitators to cessation.[25] The authors also considered support techniques provided by the SSS and found motivational interviewing to be beneficial. Interestingly, our findings did not uncover data to support a continuing relationship with a professional as important, although face-to-face opportunities did assist with initial engagement with SSSs. This is consistent with findings of research aiming to prevent premature births, where increased continuity of care had no effect on smoking cessation as a secondary outcome.[26]

## LIMITATIONS
The study had a number of limitations. These included the predominance of the white British populations served by the trial sites, meaning that barriers and facilitators may differ for population groups not included in the study. There may have been issues with delivering behavioural support when English is not the first language, or with differing cultural attitudes and behaviours related to smoking in pregnancy if a more diverse participant group had been included. In addition, resources did not permit more detailed data collection, observations and analysis of communication techniques regarding SSS communication with women.

## FURTHER WORK
CPIT III compared smoking cessation rates towards the end of pregnancy in women receiving usual care plus

financial incentives versus usual care alone. However, with usual care among sites varying substantially, it raises the question 'what influence does this variation have on the effectiveness of financial incentives?' Based on prior evidence that suggest cessation support could be a confounder for incentives,[6] the findings of this process evaluation help to understand the generalisability of the overall CPIT III trial findings. This paper was completed prior to the research team analysing or knowing quantitative trial outcomes by site. This prospective exploratory process evaluation therefore minimises any bias that could arise from retrospective analysis once trial outcomes are known. The finding that CPIT III trial sites had differing characteristics and balances of barriers and facilitators to providing smoking cessation support can now be incorporated into our a-priori proposition:

In sites with fewer barriers and more facilitators to providing cessation support, more women will stop smoking (primary outcome), engage with SSSs, be recruited to and retained in the trial. This would indicate that maternity services and SSSs potentially provide an influential contribution to the offer of financial incentives which could inform implementation decisions.

The counter-factual null hypothesis would be:

Financial incentives are effective regardless of the differing characteristics, barriers and facilitators to the provision of SSS support and usual care.

The numbers of participants taking part in individual CPIT III sites do not provide the statistical power to draw conclusions about the effectiveness of financial incentives on a local basis. However, further exploratory descriptive analysis of CPIT III results by trial site will now be conducted by the trial statistician and health economists working independently of the process evaluation team. How usual care and trial outcomes vary across sites will be important for understanding generalisability in the implementation of financial incentives.

**Author affiliations**
[1]Institute for Social Marketing and Health, University of Stirling, Stirling, UK
[2]School of Health and Life Sciences, University of the West of Scotland, Paisley, UK
[3]Department of Health Sciences, University of York, York, UK
[4]Usher Institute of Population Health Sciences and Informatics, and SPECTRUM Research Consortium, University of Edinburgh, Edinburgh, UK
[5]School of Medicine, Dentistry and Nursing, University of Glasgow, Glasgow, UK
[6]Nursing, Midwifery and Allied Health Professional Research Unit, University of Stirling, Stirling, UK

**Acknowledgements** The authors wish to thank service users, trial participants, maternity and smoking cessation staff who supported the process evaluation by taking part in formal or informal interviews, permitting observation of usual care or facilitating data collection within case study sites. Our thanks also to trial staff who provided their perspectives on trial progress locally. Finally, our thanks to Dr Isabelle Uny (PhD), Research Fellow in the Institute for Social Marketing and Health at the University of Stirling, for her assistance with early process evaluation data collection and analysis.

**Contributors** JM (MSc) led the data collection, analysis and reporting relating to the process evaluation. JM is a Research Fellow at the University of Stirling and is registered for a part time PhD by publication which is based on the Cessation in Pregnancy Incentives Trial (CPIT III) process evaluation. JM is responsible for the overall content of the manuscript as guarantor. PH and FMH designed the process evaluation for the original grant application, oversaw data collection and analysis and are PhD supervisors. LS was the trial manager of CPIT III and DMT and LB were coprincipal investigators of CPIT III and designed the overall study. All authors contributed to drafting and reviewing the manuscript.

**Funding** This study was supported by the Chief Scientist Office of the Scottish Government, grant number HIPS/16/01/CSO; with partnership funding from Cancer Research UK, grant number C48006/A20863/CRUK; Health and Social Care Northern Ireland COM/5352/17; Northern Ireland Chest Heart & Stroke 2017_09; Health and Social Care Northern Ireland Public Health Agency (no grant number); Lullaby Trust P272; Scottish Cot Death Trust (no grant number).

**Competing interests** None declared.

**Patient and public involvement** Patients and/or the public were not involved in the design, or conduct, or reporting or dissemination plans of this research.

**Patient consent for publication** Not required.

**Ethics approval** This study involves human participants and was approved by Ethical approval for CPIT III, including the process evaluation, was given by the West of Scotland Research Ethics Committee 4, reference: 17/WS/0173. Participants gave informed consent to participate in the study before taking part.

**Provenance and peer review** Not commissioned; externally peer reviewed.

**Data availability statement** No data are available. Data sharing is not possible for this study as participants were not asked for permission to share their data to avoid deterring participation, particularly from professionals working in services that may be identifiable.

**ORCID iDs**
Jennifer McKell http://orcid.org/0000-0002-2912-0837
Fiona M Harris http://orcid.org/0000-0003-3258-5624
Lesley Sinclair http://orcid.org/0000-0002-2210-8181
Linda Bauld http://orcid.org/0000-0001-7411-4260
David Michael Tappin http://orcid.org/0000-0001-8914-055X
Pat Hoddinott http://orcid.org/0000-0002-4372-9681

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
