## [Reviewer comments · BMJ Open]

ARTICLE DETAILS

TITLE (PROVISIONAL)	Usual care in a multi-centre randomised controlled trial of financial incentives for smoking cessation in pregnancy: qualitative findings from a mixed-methods process evaluation
AUTHORS	McKell, Jennifer; Harris, Fiona M.; Sinclair, Lesley; Bauld, Linda; Tappin, David; Hoddinott, Pat

VERSION 1 – REVIEW

REVIEWER	Jones, Susan Teesside University, Health and Social Care Institute
REVIEW RETURNED	08-Sep-2022

GENERAL COMMENTS	Reviewing BMJ Open 2022-066494 Usual care in a multi-centre randomised controlled trial of financial incentives for smoking cessation in pregnancy: qualitative findings from a mixed methods process evaluation McKell, J., Harris, F., Sinclair, L., Bauld, L., Tappin, D., Hoddinott, P. Overall The text clearly explains how this process evaluation sits within the CPIT III and will inform the overall findings. The underpinning theory is stated and applied in the coding framework and context, mechanism, outcomes (CMO) configurations. The completion of this work before the trial outcomes were available is a clear strength. Key research evidence has been used to justify this work. In the introduction, I like the way you have focused on the longevity of the mother, as a rationale, as focusing on the baby unhelpfully feeds into the narrative of only quitting during pregnancy. I found the way you have reported the informal discussions reassuring and refreshing, recognising as you do, how these are so important in understanding the overall picture at each site. The findings are consistent with the work completed for the process evaluation during the implementation of BabyClear© across North East England in 2013-15. This paper helpfully takes that work further, defining in detail in the CMOs the focus for attention within usual care, to maximise the effectiveness of
--

	initiatives to support women to stop smoking in pregnancy, including financial incentives. With regard to the analysis, there were a couple of gaps where I felt it would be difficult to repeat the study without further detail. I would like to see a note about how you integrated the data from different sources e.g. did you have specific codes which only received data from one type of source, or did you code all data across the framework? I would also like to see a comment on how you drew out the barriers and facilitators from the coding framework. It will be interesting to see how the trial outcomes – when complete - relate to these findings, especially regarding the variation in delivery systems and processes between sites. Please see the table below for suggested amendments:    Page Line Comment     2 7 Contextual variations in smoking cessation support are not the only factors that may affect effectiveness. Suggest amend to reflect this breadth.   2 44 Suggest amend to read “more facilitators regarding this support would be”   6 16 Please could you add the range of time for interviews? Did it vary between telephone and face to face?   6 23 Expanding on the interpretive phase: I would like a to see a note about how you integrated the data from different sources e.g. did   	Page	Line	Comment	2	7	Contextual variations in smoking cessation support are not the only factors that may affect effectiveness. Suggest amend to reflect this breadth.	2	44	Suggest amend to read “more facilitators regarding this support would be”	6	16	Please could you add the range of time for interviews? Did it vary between telephone and face to face?	6	23	Expanding on the interpretive phase: I would like a to see a note about how you integrated the data from different sources e.g. did
Page	Line	Comment														
2	7	Contextual variations in smoking cessation support are not the only factors that may affect effectiveness. Suggest amend to reflect this breadth.														
2	44	Suggest amend to read “more facilitators regarding this support would be”														
6	16	Please could you add the range of time for interviews? Did it vary between telephone and face to face?														
6	23	Expanding on the interpretive phase: I would like a to see a note about how you integrated the data from different sources e.g. did														

	you have specific codes which only received data from one type of source, or did you code all data across the framework? I would also like to see a comment on how you drew out the barriers and facilitators from the coding framework. 6 29 Suggest a hyphen to add clarity: "information-rich". 7 58 Suggest add comma: "usual care, particularly" 7 58 Suggest amend to read: "particularly in relation to professional" 8 8 It would be helpful to add the size of SSS teams, or a comment to the effect that teams are small and therefore one person off work/vacant post can have a large effect. 8 25 Please could you clarify this sentence as I was unclear what was meant by: "per protocol, in intervention participants ..." 9 33 Suggest amending to read: "if they smoked and to provide a carbon" 9 53 Suggest add comma: "among midwives, suggesting ..." 9 53 Suggest add hyphen: "taking place - unless ..." 10 34
--	---

	The CMO refers to a sense of immediacy – I wonder if it is also that start of a positive relationship, a known face, a caring person? Did you find any evidence for this? Is it supported in the literature? 12 40 BabyClear is a copyrighted name so requires © COREQ – Setting Some of this information is covered in Table 1, and some interviews were by telephone, but I would like to see where professionals were interviewed face2face and how many. And similar details for women attending the SSS (trial participants). This would also help to answer Q4 of the reviewer’s checklist. COREQ – Data collection I wondered why you decided not to append the interview schedules and suggest you add your rationale into the text. This would also help to answer Q4 of the reviewer’s checklist. CODING FRAMEWORK Please could you expand a little on how you integrated the data from different sources e.g. did you have specific codes which only received data from one type of source, or did you code all data across the framework? I would also like to see a comment on how you drew out the barriers and facilitators from the coding framework. This would also help to answer Q4 of the reviewer’s checklist.
--	---

REVIEWER	Breunis, Leonieke Erasmus MC, Obstetrics and Gynaecology
REVIEW RETURNED	18-Sep-2022

GENERAL COMMENTS	This paper provides insight in the variance between trial sites that participated in a trial examining the effect of incentives for smoking cessation during pregnancy. Based on qualitative research, the authors provide information on the context of the trial site (such as urban or rural area), on experienced barriers and facilitators for providing usual care and differences in this between trial sites. They support the evidence with statements of participants of the
--

	qualitative research. Patients and public were largely involved. I congratulate the authors on this nicely written paper. I have a few options for improvement:  1. I found it confusing that the authors write about the complete evaluation in both the abstract as in the main text but only provide results of the qualitative research. During reading I was confused about what I could expect from the paper. 2. Because the paper is about usual care, I missed some information about usual care in general in the UK. For example: is obstetric care in a hospital or in a private practice, does everybody receive this care or do women with a low socioeconomic status receive less care (and are therefore also less often referred to SSS)? 3. The authors also provided the Coreq checklist. Not all items of the checklist are described in the paper while they do matter (for example relationships with participants and data collection). I would suggest to describe them in the paper too. 4. Box 1 isn't very clear, I would suggest a table. 5. In my opinion the paragraph with the comparison with literature should have more literature supporting or contradicting the findings of this paper. The sentence that another study did not provide details about care of usual is not interesting for the readers I believe.
--	--

REVIEWER	Stacey, Tomasina King's College London, Methodologies
REVIEW RETURNED	26-Sep-2022

GENERAL COMMENTS	Thank you for giving me the opportunity to review this well written and interesting paper, which has the potential to add insight into the impact of the local context on the success of smoking cessation initiatives. It will be very interesting to see how these findings relate to the findings of the CPIT III trial. I have only very minor comments to make. P11 line 14, what does 'authentic automatic referral mean? I think this needs a little more explanation. Page 13 Line 45 – do you mean 'consistency of CO monitoring', rather than inconsistency?
---

VERSION 1 – AUTHOR RESPONSE

Reviewer 1: Mrs Susan Jones

1) Page 2, Line 7: Contextual variations in smoking cessation support are not the only factors that may affect effectiveness. Suggest amend to reflect this breadth.

Yes, we agree that contextual factors in smoking cessation support are not the only factors affecting outcomes. We have re-worded the objectives section of the abstract to avoid framing implementation contexts as the only factor affecting effectiveness but a factor for which little is known.

2) Page 2, Line 44: Suggest amend to read "more facilitators regarding this support would be

Thank you, we have added 'regarding this support' to the Conclusion section of the abstract

3) Page 6, Line 16: Please could you add the range of time for interviews? Did it vary between telephone and face to face?

We have added more information about the range of time for interviews to the Methods section and included information about how many of the interviews with professionals were face to face and how many were conducted by telephone. We looked into the duration of face to face interviews compared to telephone interviews with professionals and found that the average for both sets varied little, at around 36 minutes. A sentence detailing this has been added to the methods section also. Additionally, some further detail about the setting for face to face interviews with professionals has been added.

4) Page 6, Line 23: Expanding on the interpretive phase: I would like a to see a note about how you integrated the data from different sources e.g. did you have specific codes which only received data from one type of source, or did you code all data across the framework? I would also like to see a comment on how you drew out the barriers and facilitators from the coding framework.

Most of the qualitative data collected for the process evaluation was indexed using the entire coding framework (supplementary file) but the data presented in this paper is drawn from the theme of 'routine service characteristics' in the framework, incorporated with the theme of perspectives of usual care from interviews with trial participants. Barriers and facilitators were drawn out in the charting phase, informed by the Framework method, where matrices were used to create summaries relating to each of the case study sites. These summaries then allowed for comparison of the case study sites and the similarities and contradictions in the data. Some additional text has been added to the Analysis section of the Methods to explain more about how analysis was carried out.

5) Page 6, Line 29: Suggest a hyphen to add clarity: "information-rich".

Thank you, we have added a hyphen between 'information' and 'rich' in the Analysis section of the Methods.

6) Page 7, Line 58: Suggest add comma: "usual care, particularly"

Thank you, we have added a comma between 'usual care' and 'particularly' in the first sentence of the section 'Barriers and facilitators to providing usual care in CPIT III trial sites'

7) Page 7, Line 58: Suggest amend to read: "particularly in relation to professional

We have added this text to the first sentence in the section mentioned in comment 6 above.

8) Page 8, Line 8: It would be helpful to add the size of SSS teams, or a comment to the effect that teams are small and therefore one person off work/vacant post can have a large effect.

Thank you for highlighting this omission. Some additional text indicating the size of SSS teams has been added to this section SSS capacity to provide more information.

9) Page 8, Line 25: Please could you clarify this sentence as I was unclear what was meant by: "per protocol, in intervention participants ..."

Thank you, we agree that this statement is unclear. This sentence has been amended to clarify that carbon monoxide monitoring by SSS, at 4 weeks after a quit date, was required by the CPIT III trial protocol.

10) Page 9, Line 33: Suggest amending to read: "if they smoked and to provide a carbon"

This has now been amended in the text.

11) Page 9, Line 53: Suggest add comma: “among midwives, suggesting ...”

This has now been amended in the text.

12) Page 9, Line 53: Suggest add hyphen: “taking place - unless ...”

Thank you, this has highlighted the awkwardness of these sentences, so this has been amended further to provide greater clarification.

13) Page 10, Line 34: The CMO refers to a sense of immediacy – I wonder if it is also that start of a positive relationship, a known face, a caring person? Did you find any evidence for this? Is it supported in the literature?

No, we didn't find any evidence for this within the process evaluation and have a sentence to state this. The literature on the evidence for a continuing relationship in relation to smoking cessation outcomes has been added to the discussion.

14) Page 12, Line 40: BabyClear is a copyrighted name so requires ©

Thank you for highlighting this omission. The copyright symbol has now been added to the text.

15) COREQ – Setting Some of this information is covered in Table 1, and some interviews were by telephone, but I would like to see where professionals were interviewed face2face and how many. And similar details for women attending the SSS (trial participants). This would also help to answer Q4 of the reviewer's checklist.

Please see response to comment 3 above for information about where interviews with professionals were conducted. Regards the setting for interviews with trial participants, we can see that the previous text in the data collection section of the Methods did not indicate that all interviews with trial participants were conducted by phone:

Telephone interviews with trial participants ranged from 9 to 36 minutes.

This has now been amended to:

Interviews with trial participants were conducted by telephone and lasted on average approximately 20 minutes, with duration ranging from 9 to 36 minutes.

16) COREQ – Data collection I wondered why you decided not to append the interview schedules and suggest you add your rationale into the text. This would also help to answer Q4 of the reviewer's checklist

As in response to one of the Editor's comments, not including the topic guides in the original submission was an oversight. These have now been included as supplementary material.

17) CODING FRAMEWORK Please could you expand a little on how you integrated the data from different sources e.g. did you have specific codes which only received data from one type of source, or did you code all data across the framework? I would also like to see a comment on how you drew out the barriers and facilitators from the coding framework. This would also help to answer Q4 of the reviewer's checklist.

Please see response to comment 4 above.

Reviewer 2: Ms. Leonieke Breunis

1) I found it confusing that the authors write about the complete evaluation in both the abstract as in the main text but only provide results of the qualitative research. During reading I was confused about what I could expect from the paper.

The abstract and the introduction mention the complete evaluation to explain that the qualitative findings are part of a wider mixed methods process evaluation and to provide a foundation for the hypothesis and future work outlined in the discussion. We can see however, that this framing may be confusing. To address this, the objectives section of the abstract has been amended to place more emphasis upon the focus of the paper: variations in smoking cessation support (usual care) in sites and that this was examined qualitatively. Some information about the aims of the process evaluation featured within the introduction has also been removed to avoid distracting from the focus of the paper.

2) Because the paper is about usual care, I missed some information about usual care in general in the UK. For example: is obstetric care in a hospital or in a private practice, does everybody receive this care or do women with a low socioeconomic status receive less care (and are therefore also less often referred to SSS)?

We agree that this information is important to include. Some more detail on standard maternity care in the UK is provided.

3) The authors also provided the Coreq checklist. Not all items of the checklist are described in the paper while they do matter (for example relationships with participants and data collection). I would suggest to describe them in the paper too.

Some further items on the COREQ checklist have been added to the text in the paper including descriptions of the researchers' occupations at the time of the study; relationship with participants prior to the study; their knowledge of the researcher's role; the setting of data collection (added in response to reviewer 1), the presence of non-participants and provision of the topic guides used (added in response to the editor and reviewer 1).

4) Box 1 isn't very clear, I would suggest a table.

We agree, Box 1 isn't very clear plus there are characteristics of usual care provided in some profiles which are not presented in others. A new table 2 has been added to replace Box 1.

5) In my opinion the paragraph with the comparison with literature should have more literature supporting or contradicting the findings of this paper. The sentence that another study did not provide details about care of usual is not interesting for the readers I believe.

We accept that the sentence in the 'Comparison of findings with other studies' section referring to Moore et al's study may not be interesting to readers in its current form but this was included to make the point that some description of usual care in that study may have been important for understanding the intervention's lack of effectiveness. We prefer not to remove this sentence but have opted to outline this point better by adding a reference to a systematic review, also included in the introduction, that highlighted that cessation support alongside financial incentives may influence their effectiveness. We also agree with the need for more literature to be included in this section and have added two further references.

Reviewer 3: Dr. Tomasina Stacey

1) Page 11, Line 14: what does 'authentic automatic referral mean? I think this needs a little more explanation.

We used 'authentic' to refer to automatic referral based on the specified criteria of being a smoker and/or having a CO score above a certain threshold. We agree this is confusing, so we have added 'Adherence to automatic referral based on specified criteria' to the CMO to make clearer.

2) Page 13, Line 45: do you mean 'consistency of CO monitoring', rather than inconsistency?

No 'inconsistency of CO monitoring', was the intended term here but we realise that the way the sentence is worded suggests that all three examples used in this sentence were considered facilitators to implementation when the intention was that only the first two were facilitators and the last was a barrier. This has been re-worded to make the distinction between examples clear.

VERSION 2 – REVIEW

REVIEWER	Jones, Susan Teesside University, Health and Social Care Institute
REVIEW RETURNED	31-Oct-2022
GENERAL COMMENTS	Thank you for making the suggested amendments. I find this a very interesting paper and welcome this output from CPIT.
REVIEWER	Breunis, Leonieke Erasmus MC, Obstetrics and Gynaecology
REVIEW RETURNED	07-Nov-2022
GENERAL COMMENTS	The corrections made really did improve the manuscript!